# Points to consider in the development of national human genome editing policy

Dianne Nicol[1] , Simon Niemeyer[2], Rebecca Paxton[3] and Christopher Rudge[1,4]

[1]Centre for Law and Genetics, University of Tasmania, Hobart, TAS, Australia; [2]Centre for Deliberative Democracy and Global Governance, University of Canberra, Canberra, ACT, Australia; [3]Food Values Research Group, Faculty of Arts, University of Adelaide, Adelaide, SA, Australia and [4]Law Faculty, University of Sydney, Sydney, NSW, Australia

## Overview Review

**Keywords:**
Heritable human genome editing; non-heritable human genome editing; genome editing policy; genome editing regulation; citizen engagement

**Corresponding author:**
Dianne Nicol;
Email: Dianne.nicol@utas.edu.au

### Abstract

Clustered regularly interspaced short palindromic repeats and other genome editing technologies have the potential to transform the lives of people affected by genetic disorders for the better. However, it is widely recognised that they also raise large ethical and policy questions. The focus of this article is on how national genome editing policy might be developed in ways that give proper recognition to these big questions. The article first considers some of the regulatory challenges involved in dealing these big ethical and social questions, and also economic issues. It then reviews the outcomes of a series of major reports on genome editing from international expert bodies, with a particular focus on the work of the World Health Organization's expert committee on genome editing. The article then summarises five policy themes that have emerged from this review of the international reports together with a review of other literature, and the authors' engagement with members of the Australian public and with a wide range of experts across multiple disciplines. Each theme is accompanied by one to three pointers for policy-makers to consider in developing genome editing policy.

### Impact statement

Genome editing is a new development in science that allows precise changes to be made to DNA. There are many potential uses, including in humans to prevent or treat certain diseases and disabilities. Genome editing also raises profound ethical, legal and social concerns. A number of international expert committees have examined the risks and benefits of human genome editing. There has also been a large body of academic literature highlighting these and other concerns. Despite this, there have been few attempts to provide more specific guidance for national policymakers in developing human genome editing policy. Policy and academic literature and our own empirical work (including interviews and surveys with genome editing experts and a citizens' deliberation exercise with members of the Australian community) show that there is a range of views. However, it appears that, on balance, most of the expert reports, and most of the experts and members of the Australian public who we have engaged with suggest that there is a cautious optimism that genome editing could provide a valuable contribution to healthcare in future, provided that it is properly regulated, well researched and delivered equitably. Based on our analysis, we argue that the development of national genome editing policies should focus on five particular themes. These are (1) embedding equity and other values and principles in human genome editing policy; (2) ensuring that therapy, enhancement and other applications are appropriately regulated; (3) deciding what types of human genome editing research should be allowed and supported, recognising differing views on the status of the human embryo; (4) preparing for a future when heritable human genome editing may be shown to be safe and effective and (5) building meaningful public participation into the governance of human genome editing.

### Introduction

Therapies that make changes to the human genome have long been touted for their potential to transform the lives of people affected by genetic disorders for the better (Parrington, 2016). Yet progress has been slow, in no small part due to their uncertain safety and efficacy (Doudna, 2020). New techniques, allowing more precise and better-targeted 'editing' of the genome, have the potential to address these issues of safety and efficacy. As most readers will know already, the most promising of the new genome editing techniques is known as CRISPR (standing for Clustered Regularly Interspaced Short Palindromic Repeats). CRISPR makes use of a naturally occurring genome editing system that, together with a 'Cas' (standing for 'CRISPR-*as*sociated') enzyme, and with the aid of a 'guide RNA', can cut or 'cleave' DNA at a target site. More than any genome editing technology, the CRISPR–Cas system is widely seen as a transformative

technology, primarily because the guide RNAs that direct the CRISPR–Cas machinery to target sites can be 'programmed' to create edits more simply, reliably and effectively than any previous genome editing tool (Gaj et al., 2013; Maeder and Gerbach, 2016; Mei et al., 2016).

More recently, advanced CRISPR techniques, such as base editing (using 'CRISPR nickase' systems) and prime editing (using a fused Cas9 nuclease), have been developed to facilitate single-nucleotide alterations without introducing double-strand breaks or requiring any DNA repair template (Satomura et al., 2017). Instead, these editing techniques create 'point mutations' through the chemical alteration of a target nucleotide within a narrow editing window (Komor et al., 2016). The promise of these advanced techniques is that they will reduce the risks that have long beset CRISPR editing generally, such as 'mosaic' genetic mutations and unintended or 'off-target' modifications (Lamas-Toranzo et al., 2019; Wienert and Cromer, 2022).

Both nuclease- and nickase-based CRISPR genome editing techniques are currently being translated into clinical trial applications for adult patients. In one promising trial, haematopoietic stem cells are extracted from patients' bone marrow and then edited with CRISPR to produce high levels of foetal haemoglobin and reinfused to treat sickle-cell diseases (Vertex Pharmaceuticals, 2021; Williams and Esrick, 2021). In another recently commenced trial, immune cells (CAR T cells) are 'programmed' to attack cancerous cells by means of CRISPR base editing. In this process, nucleotide bases in donor cells are edited so that the gene for CD7 (a genetic marker in blood cancers) is changed from cytosine to thymine, thus producing a 'stop codon' that terminates the production of CD7. These edited cells are then transplanted into the patient to treat relapsed lymphoblastic leukaemia (Great Ormond Street Hospital for Children NHS Foundation Trust, 2022). While these CRISPR treatments create non-heritable changes to the genome, it is also possible to create heritable genome changes by applying the CRISPR to in vitro early-stage embryos, gametes (eggs and sperm) or germ cells that are the precursors of gametes (Baylis et al., 2020).

Despite the promise of CRISPR and other genome editing technologies, it is widely recognised that they raise large ethical and policy questions (Bubela et al., 2017; Yotova, 2017; Coller, 2019; Evans, 2020; Getz and Dellaire, 2020; Eissenberg, 2021; Evans, 2021). In response, in 2020, the World Health Organization (WHO) formed an expert panel, the WHO Expert Advisory Committee on Developing Global Standards for Governance and Oversight on Genome Editing (the 'WHO GE Committee'), to examine global responses to the increasing availability of genome editing. The Committee's work culminated in three reports: a proposed governance framework, a position paper and a set of recommendations (WHO, 2021a,b,c). It is this proposed governance framework that sets this work of the WHO expert panel apart from the many other reports on genome editing that have preceded it (Cohen et al., 2022).

Aside from treating genetic disorders, the WHO governance framework report points out that human genome editing also has other potential uses, including in the treatment of infertility, promotion of disease resistance, enhancement of human traits, improvements to robustness or quality of life and addition of non-human traits (WHO, 2021a, 6). Although speculative, this list of uses illustrates that there is no clear line between uses for therapeutic purposes, for the purpose of enhancing existing traits and the use for adding new traits.

Collectively, the WHO report and reports from a number of other expert bodies provide strong support for ongoing research into the development of CRISPR and other genome editing technologies for non-heritable purposes, subject to appropriate regulatory oversight. The reports are more circumspect, however, when it comes to heritable forms of human genome editing and research involving human reproductive cells and embryos (which the WHO refers to as 'not for reproduction germline genome editing'; WHO, 2021a, v). Nevertheless, it does tend to provide cautious in-principle support for heritable human genome editing at some stage in the future – but only after more research and community engagement, and only in limited application. To date, there has been little discussion about how to translate the work of the WHO and other expert bodies and recommendations in the academic literature into national policies on genome editing. This article attempts to contribute to filling this gap by identifying more specifically the types of matters that governments will need to consider in the development of national policy.

The article starts by outlining some of the key regulatory dilemmas relating to human genome editing. It then summarises the findings from some of the most prominent reports by expert bodies. Finally, it provides five themes for policy development. This work is based on a review of the policy and academic literature, conversations that the authors have undertaken with numerous experts in genome editing – including scientists, clinicians, ethicists, social scientists, lawyers and others – and engagement with other members of the broader community.

Specifically, in 2020–2022, the authors undertook a project in Australia, the Australian Citizens' Jury on Genome Editing. The citizen deliberation component of this project focused on the question: 'Under what conditions (or circumstances) might the application of human genome editing technology be acceptable?' A full report and an executive summary of the project have been made available to policymakers and other stakeholders and are publicly available (Nicol et al., 2022a). Details of the various forms of expert and public consultation undertaken during the course of the project are provided in these documents.

This article is shaped by the Australian Citizens' Jury on Human Genome Editing project (or Australian Citizens' Jury [ACJ]), and summarises its key policy findings. The article advances the discussion about genome editing policy beyond the general recommendations of the numerous expert opinions that have been published to date. The more targeted points for policy consideration provided in this article have been developed utilising these published works, alongside the expert opinion and community consultation exercises that were part of the ACJ project.

## Regulatory issues for human genome editing

The regulatory landscape for biomedicine is highly complex. Included in this body of laws and other forms of regulation are generalist consumer protection, privacy, anti-discrimination and tort laws; specific gene technology-related laws; regulatory approvals laws for drugs and diagnostics; intellectual property laws; internationally recognised good practice standards; professional and research guidelines ('soft law') and more. This broad landscape has been referred to elsewhere as a 'regulatory soup' (Nicol et al., 2016a).

The safety of biomedical products that are intended to be made available for therapeutic purposes is currently assessed through clinical trials. We have by now all seen that in some circumstances, the safety of a particular biomedical product can be assessed quite quickly (the prime example being the speed with

which COVID vaccines were assessed and approved; Deplanque and Launay, 2021). Usually, however, the process is far slower and can take many years. The average time for the assessment of a new drug in the United States, for example, is around 9 years (Darrow et al., 2020). Clinical trials are also expensive, and many products do not make it through to approval. While CRISPR is sometimes described as 'efficient, simple and cheap,' the cost of taking a single genome-edited product through these research and clinical trials phases will be considerable – at least many millions of dollars – and sponsor-manufacturers seek to recover these costs, and to profit from that investment, when pricing the end product (Rigter et al., 2021). As a result, unless there is government support, the price of the product makes it out of reach for most people. Issues of justice, equity and access become profound in such circumstances.

Both orthodox and heterodox models of health economics concede that the public purse is not deep enough to support all innovative treatment options; there are limits on government expenditures, even if only created by the 'real resources' in an economy (Tandon et al., 2020; Henshera et al., 2021). As a consequence, hard decisions need to be made about what treatments to prioritise. Governments around the world are recognising the need to include principles of public participation, accountability and transparency in their healthcare decision-making to ensure that public expectations are met to the largest extent possible. However, calls are being made for an even greater emphasis on new forms of democratic decision-making, with a particular focus on public participation (Nielsen et al., 2021). Given the unprecedented benefits and risks of genome editing, consideration of new models of regulation to manage these issues seems particularly apt in this context.

In the context of heritable human genome editing, the regulatory regime outlined above for the clinical trial and market approval of non-heritable human genome editing would also apply. However, heritable human genome editing raises different types of questions and is governed by additional laws, together with recurrent calls for a global moratorium (Baylis et al., 2020). The questions here are not so much about how genome editing should be undertaken, but whether it should be undertaken at all. The prospect of eliminating some of the most pernicious genetic diseases through heritable human genome editing has much appeal. Indeed, some might make the argument that there is an ethical duty to pursue heritable human genome editing to alleviate human suffering (Schleidgen et al., 2020). However, in addition to concerns about the safety of heritable human genome editing for offspring and future generations, there are significant ethical concerns that weigh against its adoption.

Without attempting to be comprehensive, some of the most commonly raised ethical concerns associated with heritable human genome editing (and research involving human embryos) include (1) the impossibility of securing consent from those most affected by the intervention (offspring and future generations); (2) the notion that these human-made interventions are akin to 'playing God' (Locke, 2020); (3) interventions undertaken for therapeutic purposes will open the door to 'designer babies' (Pieczynski and Kee, 2021) and (4) any interventions involving human embryos fail to recognise their moral status as human beings. The moral status of the embryo is the dominant concern for people of certain religious affiliations (Lee, 2022). However, ethical opposition to heritable human genome editing is of both religious and secular foundations. If heritable human genome editing is ever to become a reality, it will be necessary to find some middle ground between these conflicting ethical considerations.

Currently, the law relating to heritable human genome editing ranges across the full gamut from outright prohibition to more permissive approaches (Boggio et al., 2020). At this point in time, no country in the world has laws or other regulations that clearly allow heritable human genome editing (other than in the case of the creation of a mitochondrial DNA transplant, which may be authorised under licence in the United Kingdom and, soon, Australia) (Liu, 2020). There are some countries that do not have any relevant laws or other regulations; however, as a general rule, these countries do not have the technical capacity to perform heritable human genome editing (Baylis et al., 2020). Still, there is always a risk that it could be performed in places in the world where it is not clearly prohibited.

In 2018, He Jiankui infamously announced that he had performed heritable human genome editing on a small number of human embryos using CRISPR (Greely, 2019). Three children were born as a result (Mallapaty, 2022). The announcement sparked worldwide controversy and, according to media reports, led to the imprisonment of the scientist and two of his associates for 'knowingly violat[ing] the country's regulations and ethical principles to practice gene editing in assistive reproductive medicine' (Xinhua News, 2019). He Jiankui has since been released from prison (Browne, 2022). The controversy also led to increased attention from expert bodies about the ways in which heritable human genome editing should be regulated, as discussed below.

It is widely recognised that if heritable human genome editing is ever to become a reality, a great deal more research will need to be undertaken, including research involving human embryos. Recognising that any research involving human participants raises ethical issues, most countries have a system for ethical review and monitoring of that research. In most countries, every institution where human research is conducted must have an Institutional Review Board or Research Ethics Committee, whose task is to review and approve applications for research involving humans and monitor the progress of the research, guided by national statements on ethical conduct in research (Babb, 2020).

Genome editing of human embryos involves an increased degree of risk, as it is not possible to ascertain whether an embryo has been edited with off-target or unintended mutations without destroying the embryo in the inspection process (Yotova, 2020). Research involving human embryos also involves an additional layer of regulation in many countries, requiring any such uses to be licenced. In most countries, embryos are not allowed to be created for research purposes. The United Kingdom is one country, however, that takes a more liberal attitude; but even there, the creation of embryos for research is strictly controlled and licences are only provided to a small number of laboratories doing world-class research. In 2016, the UK Human Fertilisation and Embryology Authority granted the first authorisation for CRISPR-based genome editing on human embryos to the Francis Crick Institute in London, provided that the research focused on the first 7 days of development after fertilisation, and embryos were 'not grown past a maximum of 14 days after fertilisation' and not implanted in humans (Francis Crick Institute, 2016). Important findings are already emerging from this research. For example, it has been shown that in some instances, the changes resulting from the application of the CRISPR technique are not

always as precise as intended (Fogarty et al., 2017). These findings illustrate why it is important to be able to undertake research if the long-term goal is to allow some form of heritable human genome editing.

## Expert reports

In response to ongoing developments in genome editing, several national and international expert bodies have held meetings and produced reports addressing the many ethical, technical and legal questions posed by this new capability to make precise, targeted and predictable alterations to the human genome for therapeutic and other purposes.

The U.S. National Academies of Science, Engineering and Medicine produced one of the earliest reports considering the multifaceted ethical, legal and scientific issues raised by genome editing in 2017 (National Academies of Sciences, Engineering, and Medicine (US), 2017). The report followed an International Summit on Genome Editing held in Washington, DC in 2015. The Summit was organised jointly by the National Academies, the Chinese Academy of Sciences and the Royal Society of the United Kingdom. The report made several detailed recommendations on genome editing. Chief among them was the recommendation that basic laboratory research on genome editing should follow the existing regulatory pathways established with respect to other research on human cells and tissues.

For non-heritable human genome editing, the report recommended that existing ethical norms and regulatory infrastructure should be used to investigate and evaluate proposed interventions, that clinical trials be conducted strictly for the treatment or prevention of disease and disability and that regulators should carefully evaluate the risks and benefits for each intervention on a case-by-case basis. The report also recommended that heritable human genome editing be permitted under 'only the most compelling circumstances' (such as where couples who want to have genetically related children cannot do so by any other means) and only with ongoing, rigorous oversight and after robust public discussion (National Academies of Sciences, Engineering, and Medicine (US), 2017). A further recommendation was made against authorising clinical trials of non-heritable human genome editing or heritable human genome editing for purposes other than to treat or prevent disease or disability, indicating the need for more policy debate before enhancement-oriented applications might be permissible.

Following the 2015 International Summit on Genome Editing and the 2017 National Academies report, the Second International Summit on Human Genome Editing was held in Hong Kong in November 2018. He Jiankui's statement that he had undertaken heritable human genome editing statement was released the day before the summit commenced. The Organising Committee of the Summit released a statement in response shortly after the close of the Summit. Perhaps surprisingly, the Committee appeared to adopt a somewhat less strict approach than the National Academies in 2017, proposing that, despite wide variability in its potential risks, heritable human genome editing 'could become acceptable in the future if these risks are addressed and if a number of additional criteria are met' (National Academies of Sciences, Engineering, and Medicine (US), 2018). The Committee listed these criteria as 'strict independent oversight, a compelling medical need, an absence of reasonable alternatives, a plan for long-term follow-up and attention to societal effects'.

In 2020, the U.S. National Academies of Medicine and the National Academy of Sciences, together with the Royal Society (London), published a further report on heritable human genome editing. It was written by the International Commission on the Clinical Use of Human Germline Genome Editing, which was appointed following the He Jiankui statement (National Academy of Medicine, National Academy of Sciences (US) and Royal Society (Great Britain), 2020). No doubt in part due to the widespread and ongoing shock of the He Jiankui scandal, the language used in the report was much more circumspect than both the first report of the National Academies and the statement released by the Organising Committee of the Second Summit. The report set out 'clear and strict criteria' to be satisfied in advance of any initial uses of heritable human genome editing, and recommended extensive societal dialogue, noting that it is not possible to define any one responsible translational pathway due to the wide and variable possible applications of heritable genome editing.

Similar caution has been expressed in the European context. In March 2021, the European Commission and Directorate-General for Research and Innovation, through its European Group on Ethics in Science and New Technologies, published a report titled *EGE Opinion on Ethics of Genome Editing* (European Group on Ethics in Science and New Technologies, 2021). It acknowledged that there is '(almost) unanimous consensus' that heritable human genome editing is not safe enough for application. It made several recommendations, including the establishment of global governance initiatives, a registry for research and the creation of a platform for information sharing.

The WHO's inquiry was broader than those of most others, embracing all forms of human genome editing. In its governance framework, a raft of ethical principles were listed as relevant, including those relating to access, responsiveness, caution, broad-based participation and inclusion, fairness, social justice, non-discrimination, solidarity and global health equity. Regarding heritable human genome editing, the governance report posed questions relating to criminal and civil penalties and violations, interjurisdictional permissions, multigenerational follow-up and so on (WHO, 2021a). In general, the kinds of questions posed were rather more detailed and practical than those contemplated in previous international reports. The Committee also made a series of recommendations on the governance of human genome editing across several topics, including leadership, collaboration, research and medical travel, illegal, unethical and unsafe research, education and ethical principles. The Committee acknowledged the urgent need to build infrastructure and expertise in developing countries, thus recognising the challenges of distributing benefits to achieve global health justice in a postcolonial world. The Committee also recognised that all humans have equal moral worth, are entitled to live without a genetic disease for which there is a cure and deserve solidarity and support in pursuing and attaining positive health (WHO, 2021a).

Read together, the reports and statements of these national and international agencies lead to the conclusion that there is strong support for non-heritable human genome editing and much more cautious in-principle support for heritable human genome editing, albeit only at some stage in the future, and only in limited application. In essence, then, international science policy is leaning towards future approval of heritable human genome editing in strictly limited circumstances and subject to stringent oversight. If the recommendations included in these reports are to be adopted, it will be necessary to reform the law in many jurisdictions. What

precisely should be changed, however, is not an easy question to answer.

## Future development of policy for human genome editing

The remainder of this article summarises five policy themes that have emerged from this review of the international reports together with a review of other literature, engagement with members of the Australian public and engagement with a wide range of experts across multiple disciplines. As noted above, the full study is published elsewhere (Nicol et al., 2022a).

### Embedding values and principles in genome editing governance

Most countries have comprehensive regulatory regimes covering research involving humans, clinical trials and market approval of new therapeutic goods. Consent, efficacy, safety and clinical utility are all firmly embedded as core requirements. Although some tinkering might be required, international expert reports on genome editing suggest that these existing regulatory requirements are appropriate, at least in the context of non-heritable genome editing. There are also well-developed protections against unlawful discrimination and misuse of private information. Indeed, health technologies sit in a veritable 'soup' of regulatory requirements (Nicol et al., 2016a). Despite this, concerns have been raised in the literature that genome editing will pose particular challenges for existing regulation, leading to calls to examine its adequacy (Nicol et al., 2017).

The WHO governance framework report lists a range of values and principles that are intended to be used to describe both how governance and oversight measures should be reviewed and strengthened and what needs to be considered when they are (WHO, 2021a, 13–14). In informing how decisions are made, the report emphasises the importance of openness, transparency, honesty and accountability and the need for responsible stewardship in the context of regulation, science and research resources. A broader collection of values and principles are listed in the WHO governance framework in respect of what decisions are made.

### Equity of access

There does seem to be some consensus in the views of experts and many other citizens that the issues associated with equity of access to new healthcare are likely to be heightened in the context of genome editing as it enters into clinical practice. This is, of course, the case whenever innovative healthcare technologies first become available. We have seen this writ large during the course of the COVID pandemic, with the massive disparity in access to vaccines and treatments between the global north and the global south, and between disadvantaged and advantaged communities within countries (Wouters et al., 2021). Genome editing should not be seen as yet another tool to improve lot of the wealthy, at least in the short term.

Perhaps the most obvious issue when it comes to equity of access is the cost of treatment, exacerbated by the need for biopharmaceutical companies to recover their huge investment in research and development and the role that intellectual property plays in allowing them to do so, together with the rarity of diseases for which genome editing clinical trials are underway (Muigai, 2022). This situation may change once more genome editing therapies become available, but this is a long way off yet.

Although there are no approved treatments using genome editing at the current time, we have seen that the first gene therapy treatments have been staggeringly expensive. Zolgensma, a gene therapy for the treatment of spinal muscular ataxia, cost US$2.1 million for a one-off treatment when it first became available, making it the most expensive therapy on the market at the time. More recently, the gene therapy Hemgenix, a treatment for Haemophilia B manufactured by CSL Behring, has eclipsed that record with a price of US$3.5 million per treatment (Tanne, 2022). The tragedy for individuals with some of the most pernicious diseases is that long-awaited therapeutic advances are likely to be priced well beyond their reach, making them inaccessible without government or philanthropic support or insurance cover (Nicol et al., 2022b).

Inevitably, if governments choose to support genome editing treatments, the trade-off is that funds may need to be diverted from other healthcare imperatives in pursuit of a 'balanced budget' – often required by state or national legislation in modern economies (Humphery-Jenner, 2012). Yet, although a one-off treatment might be extraordinarily expensive, this is still worthwhile when compared with the human, social and economic costs of lifelong treatment for debilitating illnesses. The challenge of developing suitable 'quality of life' measures that take into account the whole range of relevant factors has long been discussed by scholars and policymakers for healthcare, with little progress in general (e.g., Brock, 1993). However, while the challenge remains fraught in the context of public health (see Kaplan and Hays, 2022), the emergence of 'well-being economics' in public finance policy may represent a developing shift away from a narrow economic indicator of societal progress, such as improved expenditure to Gross Domestic Product ratios (Dalziel, 2019).

### Commercialisation and intellectual property

Concerns about the commercialisation of biomedical research are not unfamiliar. There is clear evidence that the involvement of for-profit entities in genomic and stem cell research, in particular, has the propensity to undermine public trust and deter participation (Critchley and Nicol, 2011; Nicol et al., 2016b; Critchley and Nicol, 2017). This research has shown that people do understand that the for-profit sector can have a vital role to play in the translation of research into clinical products, particularly given the prolonged, costly and risky nature of clinical trials, and that there are currently no real alternatives. However, they do have expectations that the for-profit sector should not be given an entirely free rein: they should not be involved in governance and should be required to engage in benefit sharing in one form or another.

In the field of genome editing, we are already seeing a rush to patent and commercialise foundational elements of the technology, and a fierce legal battle about who owns the intellectual property rights associated with the core technology (Egelie et al., 2016). The ethical, legal and social issues raised by this rush to patent foundational genome editing tools have also been examined (Sherkow, 2017; Feeney et al., 2018). We are also seeing the emergence of more cooperative strategies, including open licencing of intellectual property and sharing of resources for genome editing research (Nielsen et al., 2018; Nicol and Nielsen, 2021). Even more interestingly, some patent holders are using intellectual property licences as a tool to foster ethical conduct (Guerrini et al., 2017), including the Broad Institute, which, through Editas Medicine Inc., are using licences that exclude ethically questionable uses, such as gene drives and germline editing (Broad Institute, 2014). As such, there is some indication of an appetite for self-regulation to ensure that the promise of genome editing is broadly shared. This has been referred

to as 'ethical governance by patent' (Sherkow et al., 2021). Notably, however, some commentators have expressed unease about leaving such decisions to patent holders rather than to democratically elected governments (de Graeff et al., 2018). After all, classical bioethical principles (e.g., beneficence) would generally require that patents should not be used to restrict access or prevent the delivery of a cure to a patient in need.

### Distributive justice

Although financial cost is one aspect of equity of access, there are also broader issues of distributive justice to consider. If human research and clinical trial recruitment favour socially advantaged groups, there is a risk that results are distorted, potentially leading to uneven outcomes and diagnoses. These and other concerns about the 'pernicious and pervasive effects of inequality' were recently addressed in an issue of *Nature* dedicated to the topic (Anon, 2022). The question of how distributive justice can be appropriately recognised and addressed is a complex and pressing issue. As Nicol has argued elsewhere (Nicol, 2021, references omitted),

> Distributive justice is failing to find traction when it comes to genome editing research and clinical trials, even in regions like Europe and North America. Individuals of European ancestry are often favoured in clinical trial recruitment, risking skewing results and potentially leading to uneven distribution of improvements in diagnosis and treatment. Ethnically and socially disadvantaged groups, scarred by past experiences, are reluctant to participate in research and clinical trials. Discussions relating to disabled people tend to focus on their medical conditions rather than their social situations. The benefits arising from these research efforts are even less likely to be available to the vast majority of the global population in the developing world, as observed from the broader experience with access to medicines.

Though finding solutions to the problems associated with the lack of distributive justice will be challenging, it is becoming ever more important than they are addressed. Otherwise, genome editing and other innovative health technologies are only likely to widen the distributive justice gap.

### Points to consider in ensuring that equity and access and other principles are embedded in genome editing policy

- As genome editing applications become available, they should be made broadly accessible to those in need, with particular a focus on those applications that alleviate human suffering, improve quality of life and reduce childhood mortality. Methods will thus need to be developed to operationalise these priorities, based on both the severity of the disorder or disability and the number of people affected, taking into account considerations broader than financial burdens and benefits.
- Intellectual property should be a tool both for facilitating the development of clinical applications of genome editing and for facilitating open and legitimate genome editing research. This may require the development of policies and guidelines to ensure that research remains unfettered by intellectual property constraints.
- Distributive justice demands both that genome editing applications are made broadly available and that research and development into new genome editing applications is targeted to those most in need domestically (and the broader global community). This will require further priority setting by governments.

### Ensuring that therapy, enhancement and other applications of genome editing to humans are appropriately regulated

The WHO governance framework report illustrates that there is considerable nuance in the ways in which genome editing can be utilised, extending beyond the simple distinction between clinical and non-clinical uses. Applications range from treatment of infertility, promotion of disease resistance and enhancement of human traits, through improvements to robustness or quality of life and the addition of non-human traits (WHO, 2021a, 6). This illustrates that there is no single 'bright line' between therapy and non-therapy, or between therapy and enhancement. This is challenging because the use of genome editing for the treatment of serious diseases and its use for enhancement purposes raise very ethical, social and policy and public concerns, as highlighted in the WHO governance framework report (WHO, 2021a, 26). As a result, the WHO has called for the strengthening of oversight measures for genome editing for enhancement (WHO, 2021a, 27).

One of the regulatory dilemmas in this area is that robust regulatory regimes already exist in different states for clinical applications of genome editing and other therapeutic goods, but these do not apply to non-clinical or non-therapeutic applications that might be considered as enhancements. There are clear options for legal redress, even in the non-clinical context, when there is actual bodily harm: for example, through criminal law and tort law. Otherwise, performing non-clinical genome editing may be regulation-free, particularly if it is performed outside of traditional models for the development and use of healthcare innovations and not by registered health practitioners. Although regulators like the U.S.-Based Food and Drugs Administration appear to have regulatory authority over the use of CRISPR kits, even for self-administration (Zettler et al., 2019), the situation is less clear in other jurisdictions.

Most innovative health technologies are complex, requiring specialist knowledge, equipment and reagents to create and administer. They, therefore, tend to be exclusively undertaken in specialist laboratories well versed in complying with best practice and manufacturing standards and regulatory requirements. By contrast, some genome editing techniques, particularly CRISPR-based technologies, are often said to be 'democratising' because they are relatively simple to use and are low-cost to purchase, especially now that the U.S.-based non-profit company Addgene makes essential CRISPR components available to general consumers at minimal cost (Montenegro de Wit, 2020). Still, some significant obstacles appear on the path to democratisation, with intellectual property and institutional barriers continuing as prominent concerns (Montenegro de Wit, 2020).

While there is much to be said for this process of democratisation, it is also likely to be accompanied by increased risk of misuse and misadventure, particularly if there is an absence of accountability and appropriate regulation. Concerns have already been raised regarding the potential outcomes of unregulated uses of genome editing, ranging from the fear of transhumanism, designer babies, bioterrorism and eugenics. These concerns are not without foundation. As noted in the WHO governance framework report (WHO, 2021a, 2),

> The Committee saw and heard evidence of challenges associated with rogue clinics, medical travel, as well as the reporting of illegal, unregistered, unethical or unsafe research and other activities including the offer of unproven so-called therapeutic interventions.

These concerns prompted the WHO expert panel to recommend the development of 'an accessible mechanism for confidential

reporting of concerns about possibly illegal, unregistered, unethical and unsafe human genome editing research and other activities' (WHO, 2021b, Recommendation 5).

Beyond this reporting mechanism, a number of jurisdictions are implementing laws to deter biohacking. For example, in 2019, the State of California enacted a law to ban the sale of CRISPR kits without a sign 'not use on one's self' and there has been at least one prosecution for the sale of improperly labelled kits (Asquer and Krachkovskaya, 2021; Mehlman and Conlon, 2021). The European Union, the UN and the WHO are also taking steps to address the risk of biohacking in the context of genetic modification.

### *Points to consider in ensuring that therapy, enhancement and other applications of genome editing to humans are appropriately regulated*

- Genome editing tools may be used in a range of ways, many of which may not yet even be contemplated. As the technology develops, it will be important for regulators to assess whether current regulatory frameworks relating to clinical products are adequate in assessing and monitoring the safety, efficacy and utility of the various uses to which genome editing tools may be put. In particular, regulators will need to focus attention on how to strengthen oversight of genome editing for enhancement purposes, in response to the call to do so by the WHO.
- Evidence of misuse of genome editing tools internationally indicates that governments should consider supporting the WHO recommendation for reporting misuse of human genome editing research and other activities. Further, governments should consider whether explicit biohacking offences should be identified and specified in legislation.

### *Deciding what types of genome editing research should be allowed and supported, Recognising differing views on the status of the human embryo*

One of the greatest points of divergence in views of participants at the Australian Citizens' Jury on Genome Editing related to the status of the human embryo and the extent to which it can be used in research aimed at assessing the viability of heritable forms of human genome editing. There are at least five different ways that human reproductive cells could be used in genome editing research. These involve:

1. using unfertilised human eggs and sperm cells or pre-embryos (prior to the first mitotic division);
2. using human embryos that are defective in some way, so that they are not suitable for use in assisted reproduction;
3. using human embryos that have been created for assisted reproduction but which are no longer needed by the people who provided the egg and sperm;
4. using embryo-like structures created not by fertilisation but derived from human pluripotent stem cells (e.g., stem cells modified to act like embryos), sometimes called blastoids, gastruloids or embryoids; and
5. using human embryos that are created by fertilisation specifically for research.

Questions relating to the status of the human embryo in research are complex and often contentious, with different layers of research, regulation, ethics and community sentiment depending on the type of use. Of the above ways to pursue embryo research, option five, involving creating embryos specifically for research, is the most controversial. However, if the goal is to investigate the safety,

efficacy and utility of heritable forms of human genome editing, it may be that only using healthy human embryos will demonstrate the safety and efficacy of a proposed intervention. In 2021, the International Society for Stem Cell Research recommended that stakeholders should 'lead a conversation on the scientific significance as well as the societal, moral and ethical issues' of allowing research using human embryos beyond 14 days or primitive streak formation (Clark et al., 2021).

Divergent views on the status of the human embryo have long been reflected in community and parliamentary debates in some countries (Nicol et al., 2022b). For instance, in Australia, 29 members of parliament voted against the recently passed legislation designed to allow licencing of mitochondrial donation, largely based on their concerns about the status of the human embryo (Nicol et al., 2022b). If legislation were ever introduced into the Australian parliament to allow the creation of human embryos for genome editing research, it seems inevitable that similar concerns will be raised. In contrast, the status of the human embryo receives much less attention in citizen deliberations in other jurisdictions (Thaldar et al., 2022).

### *Points to consider in deciding what types of genome editing research should be allowed and supported, recognising differing views on the status of the human embryo*

- Given that various non-heritable forms of genome editing research are already well underway, and clinical trials are progressing, it may not be necessary to give further consideration to how this aspect of genome editing research is regulated at the current time. However, governments may wish to consider funding priorities to ensure the interests of the public remain at the forefront of this research effort.
- It may be appropriate for governments to consider questions associated with the use of embryos for research purposes in the near future, in advance of scientific developments.
- More particularly, the creation of human embryos for research purposes remains contested. Further, public consultation is needed on such matters, including how a strict and tightly prescribed legislative regime might be created for this purpose and should this be deemed appropriate.

### *Preparing for a future when heritable human genome editing may be shown to be safe and effective*

Despite the unverified claim by He Jiankui that he has already undertaken heritable human genome editing on a number of embryos leading to live births, there is a very strong view among some members of the scientific community that this form of genome editing is far from safe and effective and must not yet be used in clinical practice. As a consequence, calls have been made for a global moratorium on all heritable human genome editing for the time being (Lander et al., 2019). Recent research involving human embryos has revealed disquieting results, perhaps lending support to the view that there should be a moratorium (Ledford, 2020). However, the call for a global moratorium has not been accepted by all scientists using this technology (Adashi and Cohen, 2019; Yua et al., 2021). Rather, many scientists call for more inclusive global governance including through 'soft tools', such as journal editors and conference organisers, international professional organisations and public and private funders.

Although these 'soft' tools are no doubt important components of the toolkit that could be used to achieve globally consistent governance, it is most unlikely that governments that have enacted

stringent prohibitory legislation (like Australia) would cede their authority to regulate in such contentious areas to bodies such as those described above. Indeed, unlike governments, these bodies can only use enforcement measures such as peer opprobrium, publishing embargos and withdrawal of funding. In a controversial and ethically fraught area such as this, it would seem far more appropriate to employ the full governance toolkit listed in the WHO governance framework report, which includes declarations, treaties, conventions, legislation and regulations; judicial rulings and ministerial decrees. After all, regulating the genome editing space is clearly not a trivial exercise.

While the scientific community and policymakers wrestle with these issues, research and clinical trials for non-heritable human genome editing continue and, where permissible, research involving human embryos is being undertaken (see, e.g., Currie et al., 2022). This means that it would be unwise to defer all discussion about how we might as a society deal with heritable human genome editing, should it be deemed safe and effective, to some time in the future. We have the present opportunity to future proof for that eventuality and this opportunity should be embraced.

Legislation allowing mitochondrial donation in the United Kingdom and Australia may provide a model for how a heritable human genome future might look (should there be considered societal, scientific and policy support for such an eventuality). In Australia, for example, the legislation specifically provides that only limited mitochondrial donation techniques can be used, and that licences must be obtained for pre-clinical research and training, clinical research and training, clinical trials and, eventually, clinical practice (Nicol et al., 2022b).

### *Points to consider in preparing for a future when heritable human genome editing may be shown to be safe and effective*

- We are not yet at a point in time in most countries where the application of heritable human genome editing to human embryos is imminent. Indeed, in most jurisdictions, it appears that we are not yet at a point in time when genome editing research using human embryos is being planned. Despite this, it is suggested that it is timely to consider the approach that might be taken, should heritable human genome editing be a realistic option for patients desiring to have healthy, genetically related children.

### **Building meaningful public participation in governance of human genome editing into the future**

Meaningful public participation should guide the way in which governance models for human genome editing are developed and reformed. Public participation requires education, engagement and other forms of capacity building with specific input from communities that are particularly affected, including people living with inherited diseases or disabilities. As a starting point, the involvement of members of the public in setting a reform agenda for genome editing is widely supported internationally, both in policy reports and academic commentary (Dryzek et al., 2020; Scheinerman, 2022). It is further recognised that the opportunity to set the agenda should not be the privilege of only scientists, medical practitioners, specialist government agencies or international committees. Instead, it should also include and be meaningfully led by members of the public.

If governments are to engage in regulatory reform, it will be crucial to engage fully with stakeholders and the public alike to facilitate inclusive and rigorous debate about the risks and benefits

of this complex and ethically fraught, but potentially transformative, scientific tool. Recent regulatory reform in Australia allowing mitochondrial donation within a strictly regulated environment was accompanied by an extensive public engagement exercise, featuring a citizens' jury, calls for public submissions, webinars, roundtables, surveys and other forms of engagement (Nicol et al., 2022b). This multimodal approach could provide a model for how public participation may be embedded in other regulatory reform proposals.

Government agencies are recognising that public participation, alongside transparency and accountability, is a pillar of good governance. These pillars become particularly important for innovative, personalised health technologies, because these interventions raise distinct scientific, ethical, legal and social issues. Although clinical trials for genome-edited products are only just underway, lessons can be drawn from the marketing approvals pathways for gene therapy products. The U.S. Food and Drugs Administration, the European Medicines Authority and the Australian Therapeutic Goods Administration are all attempting to include public participation processes in their decision-making (see Rudge et al., 2022). To date, however, such steps have been piecemeal and, in some cases, controversial (Nielsen et al., 2021).

There are increasing calls for these medicine regulators and related agencies to be more democratic in their decision-making generally. Critics have challenged regulatory authorities to move away from complicated benefit–risk calculations, including several accelerated and expedited pathways, and towards a more participatory, public model (Schwartz, 2017; Nielsen et al., 2021). The question is how to put public participation into effect. On the one hand, slavish adoption of public input could reduce the weight given to scientific evidence. On the other hand, formalisation could result in public participation requirements being applied in a tokenistic fashion. New models are urgently needed, particularly given the speed with which genome editing is being adopted in the laboratory and while promising new genome editing product leads are emerging.

### *Points to consider in building meaningful public participation in the governance of human genome editing into the future*

- Should governments decide to explore further any of the points to consider we have outlined above, public participation should be built into the process. Moreover, any new regulatory and other policy directions emerging from this exploratory process should be accompanied by ongoing public participation.
- Public participation planning should include culturally respectful inclusion of Indigenous communities.

### **Conclusion**

This article has identified five key themes for national policy development for human genome editing. As with many other forms of precision medicine, genome editing poses a number of regulatory challenges, both nationally and globally. Issues associated with equity are of paramount concerns. These include (1) equity of access to clinical applications of genome editing; (2) use of intellectual property in ways that facilitate both the development of clinical applications and openness in research and (3) distributive justice. We only need to look at the recent WHO report on accelerating access to genomics for global health (WHO, 2022) to understand why equity is such a major consideration in genomics.

What perhaps sets genome editing apart from other forms of precision medicine is the breadth of possible uses, many of which could make significant improvements to the health and welfare of individuals. This is of obvious benefit to the individual but also to society more generally. It is gratifying to see that clinical trials are currently underway for the uses of genome editing to treat some of the most pernicious human diseases. However, there is a range of other possible uses to which genome editing might be put, not all of which will have such a profound effect on people's lives. There is certainly scope in this context to consider whether some such uses should be prohibited (or regulated better).

The capacity to make heritable changes to the genome has been a primary focus of attention in the academic literature and international expert policy reports on human genome editing. It might be argued that these inquiries have disproportionately focused on the heritable potential of genome editing. Nevertheless, it is the case that the potential consequences of genome editing for future generations are profound and have not been fully canvassed at the national policy level. Currently, there are no countries that clearly allow intentional genome editing of germline cells or human embryos for implantation purposes, though some allow it for research purposes. Recent events have certainly heightened attention to ensuring that there is appropriate regulatory scrutiny of such activities in one form or another. Given that genome editing techniques continue to be refined and developed, it is necessary to consider more fully how heritable human genome editing should be regulated once safety and efficacy are assured. There needs to be more community debate about the ethical and social consequences of heritable human genome editing (and, indeed, other forms of human genome editing), to ensure that appropriate regulatory tools are in place before there is an urgent need to utilise them.

We have seen that this is a contentious area and there are diverse opinions, both among experts and within broader communities. There will be national differences in what are perceived to be appropriate uses of genome editing. This illustrates the vital importance of bringing the public into local and international conversations not just at the starting point, but as part of the ongoing process. There is a need for countries to develop clear policies, whether on precision medicine more broadly or particularly on human genome editing. The time for these discussions to commence at the national policy level is now.

**Open peer review.** To view the open peer review materials for this article, please visit http://doi.org/10.1017/pcm.2023.11.

**Data availability statement.** Data can be made available from the corresponding author on request.

**Acknowledgements.** We gratefully acknowledge the contributions of our collaborators, John Dryzek, Nicole Curato, Sonya Pemberton and Francesco Veri. We also thank the research and administrative staff and facilitators who have assisted us with our empirical work. We thank the many genome editing experts and Australian citizens whose contributions to this research, individually and collectively, have been of immense value.

**Financial support.** This work was supported by the Australian Medical Research Future Fund Genomics Health Futures Mission (Grant No. GHFMESLI000001). An Australian Research Council grant (Grant No. DP180101262) led by Nicol also informed this research.

**Competing interest.** The authors have no conflicts of interest. D.N. is chair of the Embryo Research Licencing Committee of the National Health and Medical Research Council. Her role on this project is entirely independent of this role and should not in any way be seen as representing the National Health and Medical Research Council or the Australian federal government.

**Ethics standard.** This research was approved by the University of Tasmania Human Research Ethics Committee (Project No. H0021841) and the University of Canberra Human Research Ethics Committee (Project No. 9095).

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
