## [Reviewer Report]

*Comments to Author*: Dear Editors,

Thank you for giving me the opportunity to review, “Points to Consider in the Development of National Genome Editing Policy.” It’s my opinion that the article should be accepted with minor revisions. The article is largely a summary of previous national genome editing frameworks and some description of the authors‘ current work in this regard. That’s just fine—especially for a review article, as this is. But to avoid a novelty problem, I think the article needs to articulate why a review now is necessary (many have been done previously); and, potentially, demonstrate how the principles articulated here potentially differ from previous frameworks and why those are now outdated. (This is the major advantage of a retrospective review in my opinion; telling us what we can now safely ignore.) I think they get close to making this explicit by speaking about “cautious optimism”—but they need to lay out, side by side, previous frameworks’ takes on these “points to consider,” why a recent “cautious optimism” lens is different from what came before, and, ideally, how their own work suggests different outcomes or variations of those outcomes. Without such a reframing, I’m concerned that the article lacks novelty. 

Beyond that, there are a few places where I get the sense there were vestiges of a previous article that never got off the ground—namely, spots in the article talking about “concrete” guidance for policy makers. E.g., “few attempts to provide concrete guidance for national policy makers in developing human genome editing policy”; 

“Even so, it is important to assess whether current regulatory frameworks relating to clinical products are adequate in assessing and monitoring the safety, efficacy and utility of the various uses to which genome editing tools may be put.” This review doesn’t really provide *concrete* guidance—such as whether a genome editing product should be a regulated or an exempt biological by the TGA—but, rather, “points to consider” in developing such guidance. Perhaps those sections should be edited out.

---

## [Reviewer Report]

*Comments to Author*: This article summarizes the policy and views of important international organizations in recent years on human genome editing, especially heritable human genome editing.It comprehensively analyzes the application prospects of human genome editing and many challenges it faces, such as ethics, law, social and regulatory . It affirms the benefits of human genome editing technology , and also clarifies that it should be encourage and support the research of gene editing under the strictly regulated, . Finally, five key themes for national policy development for human genome editing has been identified , aiming at promoting the sustainable, safe, healthy and rapid development of human genome editing technology in the future.

The article provides many policy-oriented suggestions for the development of human genome editing, and also strengthens the confidence of global scientists in continuous R&D. 

I have only one modification suggestion for the article content:

‘He Jiankui’, an important figure in the editing of the heritable human genome, was mentioned five times in the article. But the name were written as ‘he Jianku’, which needs to be revised;

---

## [Editor Report]

*Comments to Author*: The review article led by Nicol et al provides an overview on the national human genome editing frameworks and also authors recent work in this regard. The authors summarize the application prospects of human genome editing and challenges it faces, including ethics, law, social and regulatory. The manuscript is structured in a reasonable manner, raising five key themes for national policy development regarding human genome editing, aiming at promoting the sustainable, safe, healthy and rapid development of human genome editing technology in the future. A reasonable amount of literature is cited, and I think that this review will be useful not only for policy makers and researchers, but also for general public.

I have two major suggestions which should improve the manuscript and increase its usefulness to the reader. Once these points have been fixed, the manuscript should be acceptable for publication.

1. On page 5, it mentioned that “From media reports, it appears that three children (two in 2018 and one in 2019) may have been born as a result (Alonso and Savulescu, 2021)”. This is a fact that there are three children which was confirmed by He Jiankui himself. “May” can be deleted and another reference should be cited.

2. On theme 4.2，entitled “Ensuring that Therapy, Enhancement and Other Applications of Genome Editing to Humans Are Appropriately Regulated”, the authors discuss the regulation dilemma and points to consider regarding therapy and enhancement. Although the authors’ main focus was on therapy, I suggest that the authors should also discuss a little bit about enhancement separately. In the aspect of ethical concerns, policy regulation and public concerns, enhancement and therapy may be different.

---

## [Editor Report]

*Comments to Author*: I am satisfied with all the revisions the authors have made. The manuscript is ready for publication.